# Ability of Nicotinamide Riboside to Prevent Muscle Fatigue of Barrows Subjected to a Performance Test

**DOI:** 10.3390/metabo14080424

**Published:** 2024-07-31

**Authors:** Hanna M. Hennesy, Morgan E. Gravely, Daniela A. Alambarrio, Savannah R. Brannen, Jonathan J. McDonald, Sarah A. Devane, Kari K. Turner, Alexander M. Stelzleni, Travis G. O’Quinn, John M. Gonzalez

**Affiliations:** 1Department of Animal and Dairy Science, University of Georgia, Athens, GA 30602, USA; hanna11ma@comcast.net (H.M.H.); mgravelydavis@gmail.com (M.E.G.); alambarriod@uga.edu (D.A.A.); sbrannen1026@gmail.com (S.R.B.); jm93@illinois.edu (J.J.M.); sarah.devane.04@gmail.com (S.A.D.); kturner@uga.edu (K.K.T.); astelz@uga.edu (A.M.S.); 2Department of Animal Sciences and Industry, Kansas State University, Manhattan, KS 66506, USA; travisoquinn@ksu.edu

**Keywords:** electromyography, median power frequency, muscle fatigue, pig, root mean squared, vitamin B3

## Abstract

The objective of this study was to determine the daily dietary nicotinamide riboside (NR) dose required to maximize the delay of subjective muscle fatigue onset. Barrows (*N* = 100) were assigned to one of five treatments: a conventional swine finishing diet containing 0 (CON), 15 (15NR), 30 (30NR), 45 (45NR) mg·kg body weight^−1^·d^−1^ NR, or CON supplemented with 45 mg·kg body weight^−1^·d^−1^ NR by drench or cookie dough (DRE). All treatments were administered for the final 11 days of feeding. On supplementation d 10, barrows individually experienced a performance test at 1.09 m/s until they were subjectively exhausted. Wireless electromyography (EMG) sensors were affixed to the biceps femoris (BF), tensor fascia latae (TFL), and semitendinosus (ST) to measure real-time muscle activity. There were no treatment effects for barrow speed (*p* = 0.57), a tendency for a treatment effect (*p* = 0.07) for distance, and a treatment effect (*p* = 0.04) on time to exhaustion. Barrows of the 15NR and DRE treatments had greater (*p* = 0.05) distances to exhaustion than CON barrows but did not differ from other NR barrows (*p* > 0.11). Barrows in the 45NR treatment did not differ (*p* = 0.11) in distance from 30NR barrows but tended to have a greater (*p* = 0.07) distance compared to CON barrows. All other treatment comparisons did not differ (*p* > 0.27). Barrows in the DRE treatment moved for longer (*p* < 0.01) than CON barrows, but all other treatments did not differ from each other (*p* > 0.15). There was no treatment × period interaction for all muscles’ root mean square (RMS) values (*p* > 0.16), but there were Period effects for all muscles (*p* < 0.01) and a Treatment effect (*p* = 0.04) in the TFL. For all muscles, period 4 had greater RMS values than all other periods (*p* < 0.01), who did not differ from each other (*p >* 0.29). In the TFL, CON barrows had greater RMS values during the performance test compared to all NR treatments (*p* < 0.02), who did not differ from each other (*p >* 0.18). Overall, NR demonstrates potential in being a useful tool in fatigue prevention, but efficient administration of the compound needs further investigation.

## 1. Introduction

Fatigued pig syndrome can be used to characterize pigs arriving at the abattoir which are non-deceased, non-ambulatory, and physically non-injured but fail to keep up with their contemporaries. These pigs display fatigue signs including open-mouth breathing, muscle tremors, and increased skin reddening [1,2]. Typically, fatigued pigs recover during a 2 to 3 h lairage period after transportation; however, many pigs die before they recover [3,4]. Between 2000 and 2007, 0.44% of pigs were classified as fatigued [2] while that number increased to 0.66% by 2012 [5], indicating this is a growing issue. These studies estimated that dead and non-ambulatory pigs cost the U.S. swine industry USD 46 million in 2009, with values increasing to USD 88 million in 2015. Numerous environmental factors including the season, trailer density, sorting methods, and feed withdrawal times were studied to understand fatigue in response to stress sources [6,7,8,9]; however, biological stress indicators and mechanisms related to muscle fatigue have limited research available.

Internal substrates related to muscle fatigue include pyridine nucleotides, nicotinamide adenine dinucleotide (NAD+), and nicotinamide adenine dinucleotide + hydrogen (NADH). These nucleotides are coenzymes in muscle mitochondria which provide oxidoreductive power to the electron transport chain, which is responsible for ATP production [10,11,12]. Glycolysis production of NADH is utilized in the electron transport chain for both NADH and ATP production. In past studies, increased NAD+ levels accompanied by decreased NADH levels were observed following exercise training in both rats and canines [13,14]. The levels of NAD+ in resting muscles are seen to be greatest in mitochondria-dense fibers such as type I muscle fibers [15,16], which could indicate that the fiber type composition influences NAD+ levels. With the influence that the mitochondria concentration has on NAD+ and NADH levels, mitochondrial biogenesis or efficiency manipulation could increase potential ATP production and fatigue resistance.

Numerous studies have demonstrated that supplemental nicotinamide riboside (NR), a naturally occurring vitamin B3 analogue, positively affected varying muscle groups’ NAD+ concentrations in mice and humans [7,17,18,19,20]. Khan et al. [21] found that NR-supplemented mice predisposed to mitochondrial myopathy possessed increased mitochondrial DNA content, indicating increased energy production potential through mitochondrial biogenesis. These results are ideal, as NAD+ has a direct impact on mitochondrial ATP production within muscle [12]. Canto et al. [17] and Zang et al. [18] reported that NR-dosed mice had an increased running distance by 28 and 31%, respectively. Therefore, the objective of this study was to determine the effect of increasing the dietary NR dose on delaying the onset of subjective muscle fatigue.

## 2. Materials and Methods

The University of Georgia Institutional Animal Care and Use Committee approved the protocol used in this experiment, AUP# A2020 03-004-R2.

### 2.1. Live Animal Management

Over five replicates, finishing barrows (*N* = 100; CG36 × P26; Choice Genetics, West Des Moines, IA, USA) were individually housed in an environmentally controlled room at the University of Georgia Large Animal Research Unit (**LARU**; Athens, GA, USA). Barrows were housed in individual 5 m × 1.5 m pens with ¾ slatted and ¼ solid concrete floors. Each pen was equipped with a 2-hole dry feeder (Farmweld, Teatopolis, IL, USA) and nipple waterer to allow for ad libitum access to feed and water. Barrows arrived to the LARU prior to supplementation for an acclimation period (25 ± 11 d). Weights were taken every 7 d of acclimation to calculate ADG. For replications one to three, twenty-four h prior to 10 d supplementation, barrows were weighed, ranked from heaviest to lightest, and randomly allocated to one of two harvest groups within each two barrow strata, with group one beginning supplementation four d before group two to account for the four d separation between performance/harvest days. Within each performance/harvest group, barrows were ranked by weight and randomly assigned to an NR treatment within each five-barrow strata. During replications four and five, the same methodology was followed except performance/harvest d were a week apart.

Dietary treatments included a conventional swine finishing diet containing 0 (**CON**; negative control), 15 (**15NR**), 30 (**30NR**), or 45 (**45NR**) mg·kg body weight^−1^·d^−1^ NR (ChromaDex; Los Angeles, CA), or barrows fed the CON diet and supplemented daily 45 mg/kg NR in Karo Syrup (**DRE**; Karo; Chicago, IL, USA) for replications one to three and raw cookie dough (Nestle; Arlington, VA, USA) for replications four and five (positive control). Direct dosing methodology switched because pigs became increasingly resistant to drench dosing and required snaring for administration which caused increased stress. Feed weights were recorded at d 0, 5, and 10, and if feed needed to be added to feeders. On day 5, average daily feed intake and body weights were used to calculate the d 5 to 10NR supplementation rate. Three replicates consisted of 20 barrows and two replicates had 19 barrows with 1 barrow dying during the final two rounds from causes unrelated to supplementation.

### 2.2. Muscle Biopsy

On NR supplementation d 0, 5, and 10, the right biceps femoris (**BF**), tensor fascia latae (**TFL**), and semitendinosus (**ST**) muscles were biopsied for muscle metabolite analysis. Biopsy locations were identified with permanent marker and sterilized by removing hair using clippers (Wahl Professional, Sterling, IL, USA) and debris by aseptically scrubbing the area with concentrated betadine and applying final spray of 70% ethanol. One milliliter Lidocaine with 1% epinephrine was injected subcutaneously into each marked location. After a 10-min wait period, a 6-gauge piercing needle (Precision Needles, Denver, PA, USA) was used to create muscle tissue access for a Mammotome Elite tetherless vacuum-assisted biopsy unit (10G, 136 mm; Mammotome, Cincinnati, OH, USA) entry. Approximately 600 to 700 mg of tissue was collected from each muscle location, immediately placed in 2 mL centrifuge tubes, and frozen by submersion in liquid nitrogen. Biopsy sites were immediately cleaned with 70% ethanol and 20% betadine solution; and Alusheild (Neogen Ideal Animal Health, Lansing, MI, USA) was applied to prevent infection. After collection from all barrows, samples were immediately stored at −80 °C until further processing. The same biopsy procedure was utilized for d 5 and 10 biopsies; however, location was moved approximately 2.54 cm proximal to d 0 and 5 biopsy sites, respectively.

### 2.3. Performance Test

On supplementation d 11, barrows were randomly assigned to a test order and individually subjected to a performance test. Prior to testing, barrows were shaved using hair clippers (Wahl Professional) on the areas covering the left BF, TFL, and ST mid-belly. Shaved areas were cleaned with soapy water and air-dried, and wireless electromyography (**EMG**) electrodes (Noraxon, Scottsdale, AZ, USA) were affixed to each muscle parallel to the muscle fiber orientation utilizing KT tape (Hampton-Adams KT Athletic Tape) and livestock glue (Kamar Adhesive, Zionsville, IN, USA). Barrows were walked down a Gait4 pressure mat (GAIT4Dog walkway, CIR Systems Inc., Sparta, NJ, USA) for pre-exhaustion analysis.

During replications one to three, barrows were walked back and forth down a 16.5 m hallway by 2 handlers until subjective exhaustion was achieved. In replications four and five, one handler walked barrows around a track with a circumference of 55 m. Subjective exhaustion was determined by five stops of the barrow, not due to discomfort or distraction, which resulted in handler pressure application to rump. The same person determined subjective fatigue for all barrows and was blinded to each barrow’s treatment. Additionally, the order in which barrows were subjected to the performance test was randomly determined and not revealed to the person determining fatigue. Time was recorded for total exhaustion time and each turn at end of the hallway or trip around the track. Because exposure to the performance test was novel and some pigs are inherently resistant to handling, time spent resisting handlers or off-course was recorded. If a pig spent greater than 25% of their total movement time resisting handlers or off-course, they were removed from the analyses. Average speed was calculated as distance traveled divided by movement time, and pigs were moved at an average speed of 1.09 m/s. After the test, each barrow was walked down the Gait4 pressure mat for post-exhaustion analysis. Barrows were loaded into a transport trailer and immediately transported 90 m to the University of Georgia Meat Science and Technology Center (Athens, GA, USA) for harvest using USDA-approved methodology. After 24 h postmortem chilling, 2.54 cm cores were taken from left BF, TFL, and ST at the approximate EMG sensor location for muscle fiber type and succinate dehydrogenase (**SDH**) staining.

### 2.4. Gait4 Mat Analysis

Barrows were walked on a portable walkway system with an active area of 6.10 m in length and 0.61 m in width following the methods of Wang et al. [22] with slight modifications. Two pre- (**PRE**) and post- (**PO**) performance passes down the mat were obtained per barrow. Data evaluated included velocity, cadence, step time, step length, cycle time, and stride length. Velocity was obtained by dividing distance traveled by ambulation time (cm/s). Cadence was calculated as steps taken divided by ambulation time on the mat (steps/min). Stride length was the distance between successive ground contact of the left forelimb (cm). Cycle time was amount of time for a full stride cycle (s). Step length was measured on the horizontal axis of the walkway from the landing of the current footfall to the landing of the previous footfall on the opposite foot (cm). Stride length (cm) was calculated as the distance between the foot landing of two consecutive footfalls of the same foot (right to right, left to left).

### 2.5. Electromyography Analysis

The methods of Noel et al. [23] were followed with slight modifications for EMG analysis using a custom MATLAB Student R2022b program. Raw EMG data were processed for each electrical burst corresponding with a muscle contraction by using a 6th-order Butterworth lowpass and 6th-order highpass filters. The EMG frequency and amplitude characteristics were derived as median power frequency (**MDPF**) and root mean square (**RMS**), respectively. Data were averaged into four periods during performance test and individually normalized to first 15 s of each barrow’s run and averaged every 15 s during the performance test of each barrow. Each 15 s average of each barrow was reported and utilized for statistical analysis.

### 2.6. Serum Analyses

On supplementation d 10, prior to biopsy, barrows were restrained, and blood (12 mL) was collected from the jugular vein into a red top vacutainer tube (BD Vacutainer, Franklin Lakes, NJ, USA) for pre-sample analysis. After exsanguination, blood was collected for post-fatigue analysis. Within 10 min of collection, samples were centrifuged at 1115× *g* for 10 min at 20 °C. Serum was transferred to 2 mL microcentrifuge tubes, placed on ice, and ultimately stored at −80 °C until further analyses.

Serum samples were submitted to Clinical Pathology Lab (College of Veterinary Medicine, University of Georgia, Athens, GA, USA) for analyses of cortisol, lactate, creatine kinase (CK), and glucose levels. Cortisol was analyzed using Immulite 2000 System Analyzer (Malvern, PA, USA). Creatine kinase, lactate, and glucose were analyzed using Cobas C 31/501 Analyzer (Indianapolis, IN, USA).

### 2.7. Immunohistochemistry and Histochemistry

A 1 cm^3^ portion of each core was embedded in optimal cutting temperature (Neg-50, Epredia, Kalamazoo, MI, USA) embedding media, frozen with liquid nitrogen-cooled isopentane, and stored at −80 °C until further analysis. Two slides with two 5 µm thick cryosections per slide, collected 0.5 mm apart, for each muscle sample were collected on positively charge slides (Cardinal Health, Waukegan, IL, USA) for both fiber type and SDH analysis.

The methods of Paulk et al. [24] were followed for fiber type immunohistochemistry with slight modifications. Cryosections were incubated with blocking solution containing 5% horse serum and 0.2% TritonX-100 in phosphate-buffered saline (**PBS**). Cryosections were incubated overnight at 4 °C with a primary antibody solution containing blocking solution and 1:10 supernatant myosin heavy chain, type I, IgG2b (BAD5; Developmental Studies Hybridoma Bank, University of Iowa, Iowa City, IA, USA), 1:100 supernatant myosin heavy chain, type IIB, IgM (BF-F3; Developmental Studies Hybridoma Bank), and 1:10 supernatant myosin heavy chain, type IIA, IgG1 (SC-71; Developmental Studies Hybridoma Bank). Cryosections were washed three times with PBS for 5 min, incubated for 45 min with a secondary antibody solution consisting of blocking solution and 1:1000 Alexa-Fluor 488 goat anti-mouse IgM (Invitrogen, Waltham, MA, USA) for BF-F3, 1:1000 Alexa Fluor 594 goat anti-mouse IgG1 (Invitrogen) for SC-71, 1:1000 Alexa Fluor 633 goat anti-mouse IgG2b (Invitrogen) for BAD5, and 1:1000 wheat germ antigen Alexa Fluor 594. Cryosections were washed in PBS three times for 5 min, 5 µL of 9:1 glycerol in PBS was placed on cryosections, and they were cover-slipped for imaging.

Four representative photomicrographs were captured per sample at 40× magnification for immunohistochemistry and 10× magnification for SDH staining using a Revolve 4 Upright, Inverted, Brightfield, Fluorescent Microscope (ECHO Laboratories; Radnor, PA, USA). Immunohistochemistry results were obtained by measuring each individual muscle fiber CSA found within WGA fluoresced borders, taking the average of all CSA per muscle fiber type per image and averaging all 4 images per sample. Muscle fiber types were determined based off the color displayed under fluorescence, where Type IIB fluoresced green, Type IIA fluoresced red, Type IIX was an overlap of green and red fluorescence, and Type I fibers were shown in orange. Muscle fiber type proportions were calculated by taking the sum of all muscle fibers per image and dividing the number of each muscle fiber type total per image and averaging between the 4 images taken per sample.

### 2.8. Mitochondrial DNA Expression

Total deoxyribonucleic acids (**DNA**) were extracted from 25 mg of muscle tissue by following the manufacturer’s instructions of the Qiagen DNeasy Blood & Tissue Kit (Qiagen, Germantown, MD, USA). Ten nanograms of total DNA was amplified in triplicate using TaqMan Master Mix (Applied Biosystems, Foster City, CA, USA), the appropriate forward and reverse primers (Table 1), and TaqMan probe using a StepOnePlus Real-Time PCR System (Applied Biosystems). Thermocycler parameters were the following: initial heating at 50 °C for 2 min, denaturing at 95 °C for 10 min, 40 cycles of denaturing at 95 °C for 30 s, and annealing and extension at 60 °C for 1 min. Gene fold-change expression levels were calculated as 2^−ΔΔCt^ as reported by Livak and Schmittgen (2001). Expression was normalized to *beta-actin* expression (ΔCt) and calibrated to CON pig *mitochondrial D-loop* expression (ΔΔCt).

### 2.9. Statistics

Animal feed performance, body weight, and performance test data were analyzed as randomized complete block design with barrow as the experimental unit. Treatment (**TRT**) served as the fixed effect, and kill block served as the random effect. Muscle fiber morphometric and SDH data were analyzed with the same model but sorted by muscle. Blood metabolite and gait data were analyzed as a randomized complete block design with repeated measures. Treatment, time, and their interaction served as fixed effects, while block served as the random effect. Time served as the repeated measure with barrow as the subject and compound symmetry as the covariance structure. Electromyography data were analyzed as a randomized complete block design with a 5 × 3 factorial arrangement and repeated measures. Treatment, muscle, time, and their 2- and 3-way interactions served as fixed effects, and kill block served as the random effect. Time served as the repeated measure, with barrow as the subject and compound symmetry as the covariance structure. All data were analyzed using SAS 9.3 (SAS Institute Inc., Cary, NC, USA). Kill block was tested and found to be not different for all analyses. Pairwise comparisons between the least-squares means of the factor level comparisons were computed using the PDIFF option of the LSMEANS statement. Statistical significance was determined at *p* > 0.05, while tendencies were determined at 0.05 < *p* ≤ 0.10.

## 3. Results

There were no TRT effects for initial and final BW, pre-supplementation, and supplementation d 5 ADG, ADI, and G:F (*p* > 0.44; Table 2). There was no TRT effect (*p* = 0.37) for supplementation d 10 ADG; however, there was a TRT effect (*p* = 0.05) for ADI and a tendency for a TRT effect (*p* = 0.07) for G:F. Pigs from the 45NR treatment had reduced ADI compared to CON, 15NR, and DRE pigs (*p* < 0.05), who did not differ from each other (*p* > 0.41). Pigs from the 30NR treatment tended to have greater (*p* = 0.09) ADIs compared to 45NR pigs but did not differ from all other treatments (*p* > 0.27). Pigs from the 45NR treatment had greater G:F compared to CON, 15NR, and DRE pigs (*p* < 0.05), who did not differ from each other (*p* > 0.41). Pigs from the 30NR treatment did not differ in G:F compared to all other treatments (*p* > 0.14).

Treatment did not affect (*p* = 0.57) barrow speed, tended to affect (*p* = 0.06) distance, and affected (*p* = 0.04) time (Table 3). Control barrows moved less distance than 15NR and DRE barrows (*p* < 0.05) and tended to move less (*p* = 0.07) distance than 45NR barrows. Control and 30NR barrows did not differ (*p* = 0.27) in their distance traveled, and all NR barrows’ distance traveled did not differ (*p* > 0.11). Drench barrows moved for longer than all barrows (*p* < 0.04) except 15NR barrows (*p* = 0.15). Control, 15NR, 30NR, and 45NR barrows did not differ in their time subjected to the performance test (*p* > 0.11).

There were no TRT × time interactions for serum CK and lactate concentrations (*p* > 0.30; Figure 1). There were treatment effects for serum CK and lactate (*p* < 0.03), where CON barrows had greater CK and lactate values compared to those of all other treatments (*p* < 0.03), which did not differ from each other (*p* > 0.49). The creatine kinase PO-exhaustion concentration was greater (*p* = 0.04) compared to PRE-exhaustion CK concentration; however, there was no time effect (*p* = 0.24) for serum lactate concentration.

There were no TRT × time interactions or TRT effects for velocity, step time, cycle time, cadence, step length, or step length (*p* > 0.30; Figure 2). There was no time effect (*p* = 0.67) for step length; however, velocity, cadence, and stride length were greater PRE compared to PO (*p* < 0.05), but step time and cycle time were smaller PO (*p* < 0.05).

There were no TRT × period interactions for normalized RMS values of all three muscles (*p* > 0.16; Figure 3). There were time main effects for all muscles with normalized RMS values (*p* < 0.01). Period 4 had greater normalized RMS values than all other periods (*p* < 0.01), who did not differ from each other (*p* > 0.29). There were no TRT main effects for BF or ST normalized RMS values (*p* > 0.27), but there was a TRT main effect (*p* = 0.04) for TFL values. Values of CON pigs were greater than all other treatments’ TFL values (*p* < 0.02), but these treatments did not differ from each other (*p* > 0.18).

There were no TRT × period interactions for all muscles’ normalized MDPF values (*p* > 0.18; Figure 4). There were no time main effects for BF and ST normalized MDPF values (*p* > 0.11), but there was a time main effect (*p* < 0.01) for the TFL. Periods 1 and 2 did not differ (*p* = 0.75) from each other but had smaller values than periods 3 and 4 (*p* < 0.01), who did not differ (*p* = 0.85) from each other. There was no TRT main effect (*p* = 0.37) for the BF normalized MDPF value, but there were main effects for the ST and TFL (*p* < 0.05). In the ST, the CON normalized MDPF value did not differ from the 15NR, 30NR, and 45NR values (*p* > 0.30) but tended to be smaller (*p* = 0.06) than the DRE values. The value of 30NR was smaller than the 15NR and DRE values (*p* < 0.05) but did not differ (*p* = 0.47) from 45NR. The value of 45NR did not differ (*p* = 0.20) from 15NR but was smaller (*p* = 0.03) than the DRE value. In the TFL, the 45NR and DRE normalized MDPF values did not differ (*p* = 0.28) and were larger than the 30NR value (*p* < 0.03). Values of CON and 15NR did not differ (*p* = 0.78) from each other, nor 30NR and 45NR values (*p* > 0.16).

There were no TRT effects for all fiber types’ percentage or CSA within BF and ST (*p* > 0.20; Table 4). Within the TFL, there were no TRT effects for type I or IIB’s fiber percentages and all fiber types’ CSA (*p* > 0.31); however, there was a TRT effect (*p* = 0.03) for type IIX’s fiber percentage and a tendency (*p* = 0.09) for a TRT effect for type IIA’s fiber percentage. For both type IIA and IIX, there was no difference between the CON, 15NR, 30NR, and DRE fiber type proportions (*p* > 0.15). Treatment 45NR tended to differ (*p* = 0.07) from DRE but was greater than all other treatment groups (*p* < 0.03).

There was a treatment × muscle interaction (*p* < 0.01) for *Mitochondrial D-Loop* gene expression (Table 5). Treatment did not affect ST expression (*p* > 0.49). In the TFL, DRE barrows had greater expression than all other treatments (*p* < 0.05), who did not differ from each other (*p* > 0.37). In the BF, 45NR barrows had greater *mitochondrial D-Loop* expression than CON, 15NR, and 30NR barrows (*p* < 0.02), but did not differ (*p* = 0.60) from DRE barrows. Drench barrows had greater expression than CON and 15NR barrows (*p* < 0.02) but did not differ (*p* = 0.20) from 30NR barrows. The *Mitochondrial D-Loop* expression of 30NR barrows was greater (*p* < 0.01) than that of CON barrows but did not differ (*p* = 0.11) from 15NR barrows. The control barrow’s expression did not differ (*p* = 0.34) from 15NR expression.

## 4. Discussion

Pigs arriving at the abattoir non-deceased, non-ambulatory, and non-injured are a growing economic issue in the pork industry [2,5]. Limited research has focused on fatigued pig syndrome, especially prevention through dietary supplementation. Nicotinamide adenine dinucleotide, a muscle mitochondria coenzyme, might be key in improving fatigue resistance through its role in ATP production [12]. Numerous studies have documented that mice and human NR supplementation positively affected NAD+ concentrations and endurance [17,18,19,20]. Pig dietary NR supplementation has yet to be examined, and thus, in the current study, five NR doses were evaluated for their ability to affect mitochondrial biogenesis, fatigue resistance, and muscle fiber activity in real time.

During pre-supplementation and initial 5 d supplementation periods, barrows fed NR did not differ in ADI, G:F, or ADG. During the final 5 d of supplementation, 45NR barrows consumed 15% less feed than pigs in other treatments, which resulted in 45NR pigs having 56% greater G:F compared to CON and 15NR pigs. Commonly, NR is supplemented to humans in pill form because the product possesses a strong bitter taste. Research on pig tongue attributes indicated that this species possesses a significantly greater number of taste buds when compared to any other species [25]; therefore, decreased feed consumption may be attributed to the bitter and unappealing taste of NR. This effect did not occur until supplemented at greater than 30 mg·kg body weight^−1^·d^−1^, which means that these barrows consumed less NR than the calculated dose. Throughout all five repetitions, 20, 25, and 45% of 15NR, 30NR, and 45NR pigs did not meet intake expectations. This may dampen effects seen in past studies which saw significant biological effects when supplementing at dosages greater than 100 mg·kg body weight^−1^·d^−1^ [17,18,19,20]. From a production perspective, the G:F ratio increase indicates an improvement in growth performance through increased NR supplementation without the need for increased feed intake. With feed contributing 70% or more to pig production costs, improvement in the G:F ratio could provide producers with another incentive to supplement NR.

Fatigue resistance was tested by subjecting barrows to two different types of performance tests. The first test involved back and forth movement and was abandoned for the remainder of the trial because the pigs did not change direction smoothly, while the circular track allowed for more fluid, directional movement. When supplementing NR at 400 mg·kg body weight^−1^·d^−1^, Canto et al. [17] reported mice ran 31 and 30% further and longer, respectively. Supplementing the same dose, Zhang et al. [18] found that mice ran approximately 33% farther than control mice. In the current study, despite feed-based NR barrows consuming less NR than calculated, 15NR, 30NR, and DRE barrows moved between 27 and 42% farther than CON barrows. While all feed-based NR barrows did not differ from CON barrows in their time moving around the performance track, DRE barrows moved for 35% longer than CON barrows, which could indicate that the direct supplementation methods were the most efficient NR administration routes.

There are previous studies which utilize EMG technology in monitoring fatigue; however, there are yet to be studies utilizing EMG in NR-supplemented barrows. Electromyography sensors were attached to muscles important to pig ambulation. The current study evaluated RMS response for every 5 s of movement for each barrow during performance tests for the initial 180 s following PRE gait analysis so as to analyze all pigs for the same amount of time, which was the shortest run period. Root mean square, defined as muscle load assessment in the time domain on the basis of amplitude [26], serves as a means of evaluating muscle fiber recruitment by comparing active muscle fibers to the number utilized during initial movement [23]. During muscle contraction, MDPF from electromyography measures action potential conduction velocity or how fast action potential travels along muscle fibers [23,27]. In regard to muscle fatigue, as subjects fatigue, RMS will increase, indicating fiber recruitment, followed by a decrease, signifying fiber exhaustion, while MDPF decreases as muscles increase the repetition of muscle contraction [28,29,30].

In the current study, all muscles increased RMS by approximately 63% during the fourth period or at the end of performance test, indicating increased muscle fiber recruitment while the muscles fatigued. Utilizing the same performance test model as replications 3 to 5, Noel et al. [23] observed a 40% increase in TFL and 66% decrease in ST RMS of barrows supplemented with ractopamine. The authors concluded that the loss of active fibers in the ST resulted in the increase of active fiber recruitment in the TFL. In the BF and ST, there were no NR effects on RMS; however, 15NR, 30NR, and DRE barrow RMS values were, on average, 65% smaller compared to those of CON barrows over the entire performance period, which indicated less muscle fiber recruitment.

Noel et al. [23] found MDPF was not affected by a dietary treatment when pigs were subjected to similar performance testing. Cockram et al. [31] observed decreased ST MDPF, but TFL MDPF was not affected in sheep during prolonged walking on treadmills; however, Hagg [28] stated that typically, as muscle fatigues, MDPF decreases and MDPF differences can be accredited to animal physiology and fineness as well as performance test intensity and duration. In the current study, only the TFL had differences in MDPF where periods 3 and 4 were slightly greater than periods 1 and 2. Because of the small values differences, the biological effect remains questionable. While no NR effects were seen with BF MDPF, in the ST, 30NR barrows had approximately 33% smaller MDPF, which could be the reason these pigs’ performance test measures did not differ from those of CON barrows. In the same muscle, DRE barrows had approximately 38% greater MDPF than CON barrows, and this effect also occurred in the TFL where DRE barrows had 30% greater values. These results may indicate why DRE barrows performed better than CON barrows during the performance test.

Many studies document the effects exercise and stress elicit on multiple species’ blood metabolites. In humans, equine, and cattle, serum lactate increased as workload and stress increased [32,33,34,35]. Cortisol increased during elevated activity and stress situations due to signaling from the sympathetic nervous system’s fight-or-flight response. Studies with humans, cattle, and sheep have demonstrated increased cortisol responses when subjects were exposed to physical work or stressful situations [26,33,36]. Creatine kinase, as a response to muscle breakage, increased similarly to lactate and cortisol after exercise. In the past literature, sheep, equine, cattle, and humans all demonstrated increased serum CK after exposure to some form of physical activity [31,33,34,35]. Much research observing serum glucose and glycogen found an inverse relationship between them. In rats, after muscle contraction stimulation, glucose uptake increased while circulating glucose concentrations decreased [37,38].

In the current study, there was no NR or time effects seen in the glucose results; however, numerically, treatments CON and 15NR demonstrated decreased glucose post-performance test, translating to an increase in glucose uptake to account for muscle stimulation and energy depletion, while 30NR, 45NR, and DRE barrows had increased glucose concentrations. This numerical increase in greater-NR-dosed barrows may indicate NR’s ability to compensate for energy depletion and prevent glucose uptake compared to lesser NR dosage treatments. Cortisol measures in the current study agreed with the past literature, where after physical activity, concentrations increased; however, no differences were noted between NR doses, indicating that NR did not affect the sympathetic nervous system and its response systems during times of stress.

Serum lactate levels in the current study, contradictory to studies mentioned above, had no change between pre- and post-fatigue. During pre-fatigue blood collection, barrows resisted snaring by pulling back on the snare and thus contracting their muscles and possibly increasing the production of lactate prior to fatigue. Although lactate did not differ between pre- and post-fatigue, a 35% decrease in lactate levels was experienced in all barrows supplemented with NR compared to CON barrows. This result would indicate that NR was beneficial in decreasing overall lactate levels when pigs were subjected to the conditions associated with blood collection and performance testing. This could benefit the end-product by assisting in reducing the onset of pale, soft, and exudative pork in the industry [39].

The past literature evaluating alternative NAD+ precursors, niacin and tryptophan, on CK and lactate production has shown no effect on CK within humans, rats, dairy cattle, or pigs but has shown a decrease in lactate levels within rats, fish, and broiler chickens [40,41,42,43,44]. Similar to the previous literature, CK increased 33% due to performance testing and decreased 40% over both time periods in barrows supplemented with NR compared to CON barrows. These results suggest that NR prevented activity-catalyzed muscle damage, most likely because of reduced serum lactate accumulation. Because CK serves as a key enzyme in ATP production [45], this result could also indicate that there was more CK present in the muscle to allow NR pigs to perform for longer.

Although gait mat analysis showed no NR effect, all parameters, except stride length, showed differences pre- and post-fatigue. Previous studies comparing fatigued to non-fatigued human gait profiles showed decreases in stride length and cadence and increases in step and cycle time in fatigued subjects [46,47,48]. Similarly, in the current study, pre- to post-performance testing showed 20, 12, and 10% decreases in velocity, cadence, and stride length, respectively, and 17 and 14% increases in step time and cycle time. These noticeable differences between the two time points indicate the potential usefulness of gait mat analyses in production settings to flag fatigued pigs. Human therapists are introducing gait mat usage into rehabilitation programs following treatment to evaluate the efficacy of treatments [48]. As is being done in human medical facilities, the inclusion of gait mats at farms and abattoirs may increase fatigued pig awareness. When pigs are flagged as possibly fatigued, they may be allowed an increased lairage period or shorter distance to holding pens in order to minimize transport loss due to fatigued pigs.

Over all muscles and barrows, there was 13% Type I, 23% Type IIA, 12% Type IIX, and 52% Type IIB muscle fibers. These distributions were similar to those reported by Noel et al. [23] in pig ST, TFL, triceps brachii lateral head, and deltoideus. There were no treatment effects on BF or ST fiber type distribution; however, within the TFL, there were greater proportions of Type IIX fibers in 45NR barrows compared to CON, 15NR, and 30NR barrows. Although it is difficult to prove, these findings may indicate a transition to more mitochondrial dense Type IIA fibers. A longer supplementation period could possibly allow for a full transition of Type IIX fibers to either IIB or IIA to establish a clearer effect NR may have on fiber types, as muscle fibers take at least 2 weeks, and possibly longer, to fully turnover [49]. Although it is difficult to prove, the current study’s 45NR barrows tended to have a greater number of Type IIA fibers, which would indicate a shift towards a more oxidative fiber type. As seen in Noel et al. [23], the current study, although not statistically analyzed, showed increased CSA going from Type I to Type IIB; this is expected, as the past literature has established an inverse relationship between oxidative capacity and CSA [50]. However, there was no response noted between treatments for CSA.

While fiber type could indicate changes in muscle metabolism, direct measures such as mitochondrial DNA content serve as a better indicator. Khan et al. [21] demonstrated that 400 mg/kg BW of NR increased mitochondria biogenesis in mice with a mitochondrial myopathy. When supplemented to a developing chicken embryo, NR increased hatched chick mitochondrial expression by 74% [51]. In the current study, all NR barrows’ BF mitochondrial DNA expression increased from 0.24- to 1.14-fold compared to that of CON barrows. In the TFL, only DRE barrows had 0.05-fold greater mitochondrial expression than CON barrows. These results indicate increased mitochondrial biogenesis could have been responsible for the increased performance of NR pigs, and DRE pigs performed the best, possibly because of the increased mitochondrial expression.

## 5. Conclusions

Feeding barrows a formulated diet containing varying dosages of NR for 10 d increased fatigue resistance in a dose-dependent manner, with direct supplementation providing the best results. There was no treatment effect on the fiber type composition of the muscles analyzed, but EMG analyses indicated NR barrows had less muscle fibers recruited during testing. Blood metabolite analyses also indicated that NR pigs had less serum lactate accumulation, which may have resulted in reduced serum CK accumulation, both indicators of less fatigue. All NR barrows had greater BF mitochondrial DNA expression, but DRE barrows were the only barrows to have increased expression in the TFL, which could also be a contributing factor to their increased performance during testing. Overall, these results indicate that NR serves as a promising countermeasure to muscle fatigue, but the route of administration needs to be improved.

## Figures and Tables

**Figure 1 metabolites-14-00424-f001:**
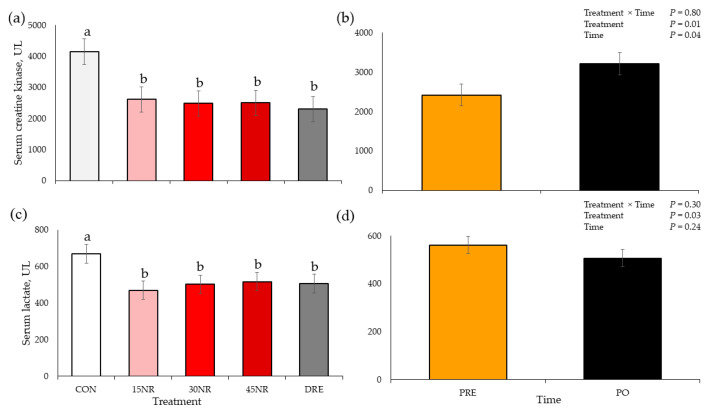
Serum creatine kinase (**a**) treatment and (**b**) time main effects and serum lactate (**c**) treatment and (**d**) time main effects of pigs pre- and post-performance test following 10 d of feeding a conventional swine finishing diet containing 0 (**CON**), 15 (**15NR**), 30 (**30NR**), 45 (**45NR**) mg·kg body weight^−1^·d^−1^ NR, or barrows being supplemented daily 45 mg/kg body weight NR in Karo Syrup (**DRE**) for repetitions 1 to 3 and raw cookie dough for repetitions 4 and 5. ^a,b^ Means within a panel with different superscripts differ (*p* < 0.05).

**Figure 2 metabolites-14-00424-f002:**
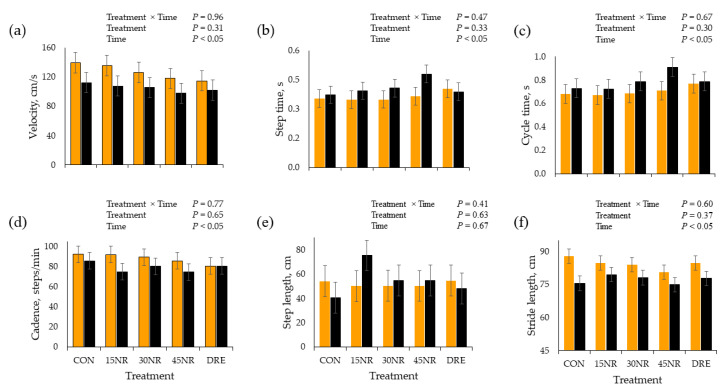
Gait4 Mat averaged (**a**) velocity, (**b**) step time, (**c**) cycle time, (**d**) cadence, (**e**) step length, and (**f**) stride length of barrows pre- (yellow) and post-performance test (black) following 10 d of feeding a conventional swine finishing diet containing 0 (**CON**), 15 (**15NR**), 30 (**30NR**), 45 (**45NR**) mg·kg body weight^−1^·d^−1^ NR, or barrows supplemented daily 45 mg/kg NR in Karo Syrup (**DRE**) for repetitions 1 to 3 and raw cookie dough for repetitions 4 and 5.

**Figure 3 metabolites-14-00424-f003:**
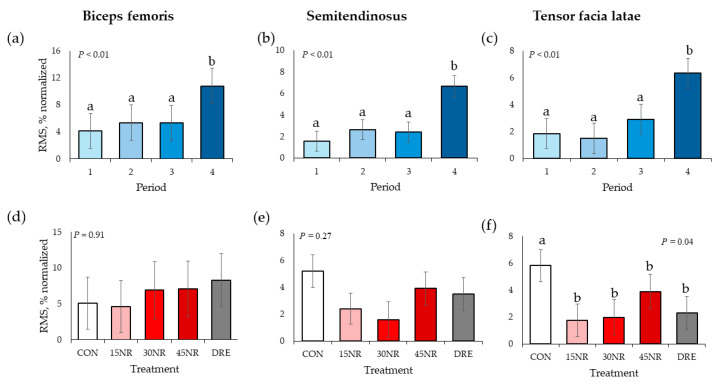
Electromyography root mean square (**RMS**) by period for barrow (**a**) bicep femoris, (**b**) semitendinosus, and (**c**) tensor facia latae and by treatment for the (**d**) bicep femoris, (**e**) semitendinosus, and (**f**) tensor facia latae during performance testing following 10 d of feeding a conventional swine finishing diet containing 0 (**CON**), 15 (**15NR**), 30 (**30NR**), 45 (**45NR**) mg·kg body weight^−1^·d^−1^ NR, or barrows being supplemented daily 45 mg/kg body weight NR in Karo Syrup (**DRE**) for repetitions 1 to 3 and raw cookie dough for repetitions 4 and 5. ^a,b^ Means within a panel with different superscripts differ (*p* < 0.05).

**Figure 4 metabolites-14-00424-f004:**
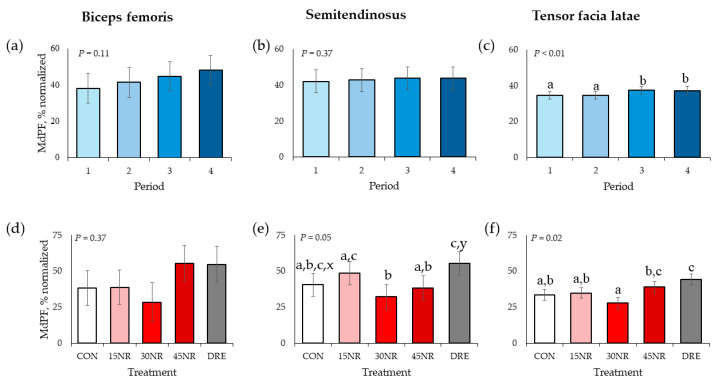
Electromyography median power frequency (**MdPF**) by period for barrow (**a**) bicep femoris, (**b**) semitendinosus), and (**c**) tensor facia latae and by treatment for the (**d**) bicep femoris, (**e**) semitendinosus, and (**f**) tensor facia latae during performance testing following 10 d of feeding a conventional swine finishing diet containing 0 (**CON**), 15 (**15NR**), 30 (**30NR**), 45 (**45NR**) mg·kg body weight^−1^·d^−1^ NR, or barrows being supplemented daily 45 mg/kg body weight NR in Karo Syrup (**DRE**) for repetitions 1 to 3 and raw cookie dough for repetitions 4 and 5. ^a,b,c^ Means within a panel with different superscripts differ (*p* < 0.05). ^x,y^ Means within a panel with different letters tend to differ (*p* ≤ 0.10).

**Table 1 metabolites-14-00424-t001:** Gene-specific primers utilized for qPCR analysis.

Gene	Forward Primer, 5′-3′	Reverse Primer, 5′-3′	T*_w_* ^1^	Amplicon Length, bp	Efficiency	GeneBank Accession
*Beta-actin*	CCCCTCCTCTCTTGCCTCTC	AAAAGTCCTAGGAAAATGGCAGAAG	59.1	73	90	DQ452569.1
Probe	6FAMTGCCACGCCCTTTCTCACTTGTTCTMGBNFQ					
*Mitochondrial D-loop* ^2^	GATCGTACATAGCACATATCATGTC	GGTCCTGAAGTAAGAACCAGATG	55.0	197	93	OR237429.1
Probe	6FAMCCAGTCAACATGCGTATCACCACCAMGBNFQ					

^1^ Melting temperature, °C; ^2^ Normalization gene not affected (*p* > 0.10) by treatment.

**Table 2 metabolites-14-00424-t002:** Pre-supplementation and supplementation d 5 and d 10 body weights and feed performance of barrows fed a conventional swine finishing diet containing 0 (**CON**), 15 (**15NR**), 30 (**30NR**), 45 (**45NR**) mg·kg body weight^−1^·d^−1^, and 45 mg/kg NR daily oral drench (**DRE** ^1^) nicotinamide riboside.

Item	Treatment						
	CON	15NR	30NR	45NR	DRE	SEM	*p*-Value
Finishing performance							
Initial BW, kg	95	94	95	95	96	5.1	0.96
Final BW, kg	134	133	134	133	133	2.9	0.99
Pre-supplementation ^2^							
ADG, kg	1.01	1.07	1.01	0.99	0.95	0.07	0.60
ADI, kg	3.56	3.48	3.49	3.38	3.51	0.22	0.72
G:F	0.29	0.33	0.30	0.31	0.28	0.03	0.51
D5 supplementation ^3^							
ADG, kg	1.65	1.44	1.41	1.55	1.53	0.16	0.44
ADI, kg	3.64	3.41	3.56	3.52	3.57	0.23	0.74
G:F	0.46	0.46	0.41	0.46	0.45	0.06	0.93
D10 supplementation ^4^							
ADG, kg	2.89	2.41	2.69	2.51	2.24	0.31	0.37
ADI, kg	3.74 ^a^	3.68 ^a^	3.50 ^a,b,x^	3.14 ^b,y^	3.56 ^a^	0.20	0.05
G:F	0.25 ^a^	0.30 ^a^	0.34 ^ab^	0.43 ^b^	0.29 ^a^	0.05	0.07

^1^ Nicotinamide riboside administered in Karo Syrup (Karo; Chicago, IL, USA) for periods 1 to 3 and raw cookie dough (Nestle; Arlington, VA, USA) for periods 4 and 5. ^2^ Performance parameters for barrows prior to NR supplementation. ^3^ Performance parameters for barrows during first 5 d of supplementation. ^4^ Performance parameters for barrows during final 5 d of supplementation. ^a,b^ Means within a row with different letters differ (*p* ≤ 0.05). ^x,y^ Mean within a row with different letters tend to differ (*p* ≤ 0.10).

**Table 3 metabolites-14-00424-t003:** Performance test and pre- and post-performance testing blood parameters of barrows fed a conventional swine finishing diet containing 0 (CON), 15 (**15NR**), 30 (**30NR**), 45 (**45NR**) mg·kg body weight^−1^·d^−1^, and 45 mg/kg NR daily oral drench (**DRE**
^1^) nicotinamide riboside.

	Treatment						*p*-Value		
Item	CON	15NR	30NR	45NR	DRE ^1^	SEM	Treatment	Time	Treatment × Time
Performance test ^2^									
Average speed, m/s	1.02	1.10	1.13	1.11	1.07	0.69	0.57	-	-
Exhaustion distance, m	415 ^a,x^	531 ^b^	485 ^a,b^	526 ^a,b,y^	588 ^b^	47.4	0.07	-	-
Exhaustion time, s	420 ^a^	498 ^a,b^	452 ^a^	461 ^a^	569 ^b^	42.2	0.04	-	-
Blood parameter									
Serum cortisol, µg/dL							0.85	<0.01	0.99
Pre ^3^	3.62	3.50	3.91	2.69	3.58	0.91			
Post ^4^	14.17	13.54	13.88	13.33	13.49	0.86			
Serum glucose, mg/dL							0.30	0.94	0.32
Pre	86.98	84.89	87.22	86.93	86.83	5.51			
Post	77.60	77.75	93.05	90.21	93.16	5.06			

^1^ Nicotinamide riboside administered in Karo Syrup (Karo; Chicago, IL, USA) for periods 1 to 3 and raw cookie dough (Nestle; Arlington, VA, USA) for periods 4 and 5. ^2^ Barrows were individually walked at 1.09 m/s until subjective exhaustion occurred. Exhaustion was determined as the barrow resisting movement and resisting human application of pressure to rump 5 times. If barrows went down on front limbs, exhaustion was also designated. ^3^ Blood was collected on d 10 of supplementation prior to performance test. ^4^ Blood was collected immediately following stunning after performance test. ^a,b^ Means within a row with different letters differ (*p* ≤ 0.05). ^x,y^ Mean within a row with different letters tend to differ (*p* ≤ 0.10).

**Table 4 metabolites-14-00424-t004:** Immunohistochemistry muscle fiber characteristics of three muscles in barrows fed a conventional swine finishing diet containing 0 (CON), 15 (**15NR**), 30 (**30NR**) 45 (**45NR**) mg·kg body weight^−1^·d^−1^, and 45 mg/kg NR daily oral drench (**DRE**
^1^) nicotinamide riboside.

Item	Treatment ^1^					SEM	*p*-Value
	CON	15NR	30NR	45NR	DRE		
Fiber type, %							
*Biceps femoris*							
Type I	0.15	0.14	0.17	0.14	0.14	0.01	0.36
Type IIA	0.22	0.27	0.26	0.27	0.24	0.02	0.58
Type IIX	0.11	0.08	0.07	0.08	0.09	0.02	0.66
Type IIB	0.52	0.51	0.49	0.50	0.53	0.02	0.83
*Semitendinosus*							
Type I	0.07	0.08	0.08	0.07	0.12	0.02	0.22
Type IIA	0.20	0.19	0.15	0.24	0.24	0.03	0.29
Type IIX	0.16	0.13	0.16	0.16	0.11	0.03	0.46
Type IIB	0.57	0.60	0.61	0.53	0.53	0.03	0.35
*Tensor facia latae*							
Type I	0.14	0.16	0.17	0.16	0.15	0.02	0.88
Type IIA	0.27 ^a^	0.26 ^a^	0.27 ^a^	0.20 ^b^	0.22 ^a,b^	0.03	0.09
Type IIX	0.12 ^a^	0.09 ^a^	0.11 ^a^	0.18 ^b^	0.13 ^a,b^	0.02	0.03
Type IIB	0.52	0.51	0.49	0.50	0.53	0.02	0.83
Cross-sectional area, µm^2^							
*Biceps femoris*							
Type I	2753	3049	3019	3102	3112	299	0.90
Type IIA	3917	4392	4440	4575	4146	393	0.78
Type IIX	5148	5476	6335	5894	5674	558	0.64
Type IIB	5775	6524	6662	6258	6260	624	0.70
*Semitendinosus*							
Type I	3583	4433	4252	4507	3603	515	0.47
Type IIA	4338	4796	4980	6098	4668	606	0.20
Type IIX	5264	6238	6231	6840	6589	759	0.55
Type IIB	4589	5801	5463	5997	6260	805	0.46
*Tensor facia latae*							
Type I	1879	1698	2059	1849	2085	249	0.74
Type IIA	2324	2455	3243	2388	2753	341	0.31
Type IIX	3263	3450	4282	3459	3839	409	0.43
Type IIB	3890	3912	4661	4241	4367	394	0.57

^1^ Drench treatment nicotinamide riboside administered in Karo Syrup (Karo; Chicago, IL, USA) for periods 1 to 3 and raw cookie dough (Nestle; Arlington, VA, USA) for periods 4 and 5. ^a,b^ Means within a row with different letters differ (*p* ≤ 0.05).

**Table 5 metabolites-14-00424-t005:** *Mitochondrial D-Loop* expression of barrows fed a conventional swine finishing diet containing 0 (**CON**), 15 (**15NR**), 30 (**30NR**), 45 (**45NR**) mg·kg body weight^−1^·d^−1^ nicotinamide riboside (**NR**), and 45 mg/kg body weight NR daily by oral drench (**DRE**).

Item	Treatment						*p*-Value		
	CON	15NR	30NR	45NR	DRE ^1^	SEM	Treatment	Muscle	Treatment × Muscle
Expression, fold							<0.01	<0.01	<0.01
*Semitendinosus*	0.84	0.65	0.73	0.76	0.75	0.240			
*Tensor facia latae*	1.00 ^a^	0.75 ^a^	0.88 ^a^	0.91 ^a^	1.54 ^b^	0.231			
*Biceps femoris*	0.96 ^a^	1.2 ^a,b^	1.64 ^b,d^	2.28 ^c^	2.09 ^c,d^	0.314			

^1^ Drench treatment nicotinamide riboside administered in Karo Syrup (Karo; Chicago, IL, USA) for periods 1 to 3 and raw cookie dough (Nestle; Arlington, VA, USA) for periods 4 and 5. ^a,b,c,d^ Means within a row with different letters differ (*p* ≤ 0.05).

## Data Availability

The raw data supporting the conclusion of this article will be made available by the authors, without undue reservation.

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
