# Peer review of "Ability of Nicotinamide Riboside to Prevent Muscle Fatigue of Barrows Subjected to a Performance Test"

_metabolites, 2024, doi:10.3390/metabo14080424_

Round 1

Reviewer 1 Report

Comments and Suggestions for Authors

Dear authors,

This is an interesting, important, and very little studied  subject. There are some minors issues that should be corrected but the authors must add information on the eventual difference in DRE treatments between blocks. It is not clear in the current version of the manuscript if changing from drench to cookie dough had an impact in the results. The authors have to present a statistical analysis showing these two sources did not interfere with the results. Otherwise, mixing these two sources of NAD+ together makes this treatment is very questionable.

Please, find enclosed all comments and suggestions in the PDF file.

Thank you

Comments on the Quality of English Language

Overall a good writting and language use. Minor editing issues to be corrected.

Author Response

On behalf of the authors on this publication, I want to thank you and the reviewers for taking the time to review our manuscript. Within the manuscript document, all changes have been denoted in blue text. We have completed all in-text suggestions and addressed line specific questions below. Please let me know if you have any other questions.

Reviewer 1:

Lines 6-16: If authors come from the same institution and department, it should be described only once. Describe email separately.

This is the format the journal asked for us to use so we will let them make any copy editing decisions. But yes, I agree with your statement.

Line 18: Fatigue is a very wide term. Precise what kind of fatigue.

The word “muscle” has been added here and in the title of the manuscript.

Lines 105-106: Why such change? Was there any statistical analysis to check for differences between these replicates?

We switched method of “drench” supplementation because it became increasingly stressful on the pigs to dose them with the syrup. Pigs became resistant to the dosing so we were having to snare them which caused much stress to these pigs which was a factor that other pigs were not experiencing. Therefore, we switched to the cookie dough method. A brief explanation has been added.

Line 258: Considering the different treatments offered between block 1,2 and 3 vs block 4 and 5, were block tested for difference? If so, it should be presented in the results so readers will be assured that the difference in treatments did not interfere with the overall results.

In the statistical analyses adding the kill block as random effect accounts for the variation that may have occurred; however, block was tested as a main effect and not found to be significant. Information added to line 268

.

Table 3: Editing issue

Corrected. Thank you!

Table 3: Editing issue

Corrected. Thank you!

Figure 2: It should be better indicated which colors correspond to pre- and post-performance test.

Colors denoted in legend.

Table 5: This presentation suggests that semitendinosus muscles was also affected by treatments. Please, make the appropriate change for accuracy.

The reviewer is incorrect in their assessment of the table. The interaction and main effect P-values are presented in the table. Because the interaction is significant, we chose to compare treatments within each muscle. Because the ST does not have any superscripts, that tells the reader there are no treatment effects within this muscle.

Lines 439-440: Which of the studied parameters indicates NAD+ production?

Reviewer is correct. Sentence has been rewritten.

Line 463: editing issue

“father” corrected to “further”. Thank you!

Line 590: Editing issue

Line removed. Thank you!

Lines 599-600: There seems to be an issue with this sentence. Please, check.

Sentence has been edited. Thank you!

Reviewer 2 Report

Comments and Suggestions for Authors

The provided article discusses a study on the ability of nicotinamide riboside (NR) to prevent fatigue in barrows subjected to a performance test. Here are the areas where the author needs to improve in the manuscript:

1. The study used a relatively small sample size of 100 barrows. While this may be sufficient for the statistical analysis performed, a larger sample size could provide more robust results and reduce the likelihood of type I or type II errors.

2. The study was conducted on a specific breed and age group of pigs. It is unclear how these findings would translate to other breeds, younger or older animals, or different farming conditions.

3. The study compared different methods of NR administration. However, the article does not deeply explore the bioavailability or the most efficient method of NR delivery, which could impact the effectiveness of the supplement.

4. The article does not mention whether the study was blinded. Blinding is important to reduce bias in experimental studies, especially when subjective measurements like fatigue are involved.

5. This study suggests that NR may improve fatigue resistance. Suggest the author to conduct in-depth research or discussion on the underlying mechanisms at the cellular or molecular level.

Author Response

On behalf of the authors on this publication, I want to thank you and the reviewers for taking the time to review our manuscript. Within the manuscript document, all changes have been denoted in blue text. We have completed all in-text suggestions and addressed line specific questions below. Please let me know if you have any other questions.

  1. The study used a relatively small sample size of 100 barrows. While this may be sufficient for the statistical analysis performed, a larger sample size could provide more robust results and reduce the likelihood of type I or type II errors.

We do not disagree with this assessment but to conduct the type of analyses presented here on thousands of pigs will be impossible from a cost and time standpoint. Obviously, we seek to prove this countermeasure in future studies utilizing industry measures.

  1. The study was conducted on a specific breed and age group of pigs. It is unclear how these findings would translate to other breeds, younger or older animals, or different farming conditions.

That is unknown but beyond the scope of this manuscript. We hope this is the first of many papers utilizing this compound.

  1. The study compared different methods of NR administration. However, the article does not deeply explore the bioavailability or the most efficient method of NR delivery, which could impact the effectiveness of the supplement.

Again, a valid point but beyond the scope of the study. As written, there is indication the direct dose methods are the most efficient based on performance test results. Bioavailability could be explored in future studies.

  1. The article does not mention whether the study was blinded. Blinding is important to reduce bias in experimental studies, especially when subjective measurements like fatigue are involved.

Excellent point! Information has been added that the same person (me) who was blinded to treatments determined subjective fatigue.

  1. This study suggests that NR may improve fatigue resistance. Suggest the author to conduct in-depth research or discussion on the underlying mechanisms at the cellular or molecular level.

This request is beyond the scope of this study and to hypothesize on mechanisms would be a disservice to the manuscript. We respectfully decline this request but plan to conduct future studies.

Round 2

Reviewer 1 Report

Comments and Suggestions for Authors

Dear authors, 

All comments were properly answerd.

Thank you